# Germination Promotes Flavonoid Accumulation of Finger Millet (*Eleusine coracana* L.): Response Surface Optimization and Investigation of Accumulation Mechanism

**DOI:** 10.3390/plants13162191

**Published:** 2024-08-08

**Authors:** Jing Zhang, Jia Yang, Yongqi Yin

**Affiliations:** 1College of Food Science and Engineering, Yangzhou University, Yangzhou 225009, China; mz120222090@stu.yzu.edu.cn; 2Yangzhou Center for Food and Drug Control, Yangzhou 225000, China

**Keywords:** flavonoid, *Eleusine coracana* L., germination, response surfaces method, accumulation mechanism

## Abstract

Germination is an effective measure to regulate the accumulation of secondary metabolites in plants. In this study, we optimized the germination conditions of finger millet by response surface methodology. Meanwhile, physiological characteristics and gene expression were measured to investigate the mechanism of flavonoid accumulation in finger millet at the germination stage. The results showed that when germination time was 5.7 d, germination temperature was 31.2 °C, and light duration was 17.5 h, the flavonoid content of millet sprouts was the highest (7.0 μg/sprout). The activities and relative gene expression of key enzymes for flavonoid synthesis (phenylalanine ammonia-lyase, 4-coumarate-coenzyme a ligase, and cinnamate 4-hydroxylase) were significantly higher in finger millet sprouts germinated at 3 and 5 d compared with that in ungerminated seeds (*p* < 0.05). In addition, germination enhanced the activities of four antioxidant enzymes (catalase, peroxidase, superoxide dismutase, and ascorbate peroxidase) and up-regulated the gene expression of *PAL* and *APX*. Germination increased malondialdehyde content in sprouts, which resulted in cell damage. Subsequently, the antioxidant capacity of the sprouts was enhanced through the activation of antioxidant enzymes and the up-regulation of their gene expression, as well as the synthesis of active substances, including flavonoids, total phenolics, and anthocyanins. This process served to alleviate germination-induced cellular injury. These findings provide a research basis for the regulation of finger millet germination and the enhancement of its nutritional and functional properties.

## 1. Introduction

Finger millet (*Eleusine coracana* L.) is one of the three major cereal crops in the arid zones of the world, native to Africa and later introduced to cultivation in various parts of India and Asia [1]. As an adaptable crop, finger millet can be grown in a variety of harsh conditions [2]. Finger millet is rich in vitamins, sulfur-containing amino acids, dietary fiber, phenolic acids and flavonoids, and other nutrients and active substances [3]. Although finger millet has excellent nutritional and functional properties, it is not highly utilized, mainly due to the hard seed coat, which has a rough texture after crushing, is difficult to form a gluten network structure with, and has poor processing and molding properties [4]. Moreover, there are many anti-nutritional factors, such as phytic acid, polyphenols, and tannins, in finger millet that can inhibit the bioavailability of nutrients to the human body [5]. These unfavorable conditions greatly limit the acceptability and commercial development of the processed products of finger millet.

Germination, as a traditional, simple, and economical processing method, is effective in enhancing the content and utilization of nutrients in many cereal seeds. It has been found that germination increases the micronutrient content of finger millet, enhances protein digestibility, and, most importantly, reduces the amount of some of the anti-nutritional factors contained in the seeds [6,7]. Moreover, the grain structure is softened after germination, and the sensory flavor is improved when added as a raw food ingredient, which gives finger millet the potential for more applications in food additives [8]. In past studies on plant germination, it has been found that the content of secondary metabolites in many plants is significantly enhanced after germination, e.g., germination increased the content of phenolic acids in barley [9], isoflavones in soybeans [10], and GABA in cereals [11]. Many active substances, such as polyphenols, flavonoids, and alkaloids, also increase gradually after germination of finger millet [12,13]. These active substances in plants have become a hot research topic in recent years because of their unique functional properties that are beneficial to the human body. Flavonoids have received much attention in recent years as a common plant secondary metabolite. It is thought to be a class of natural antioxidants that exhibit significant antioxidant activity, anti-aging, and anti-inflammatory effects in humans [14]. Germination involves biochemical changes and transformations, leading to the synthesis of secondary metabolites as a defense mechanism against biotic and abiotic stressors. Flavonoids, as important plant secondary metabolites, enhance plant stress tolerance [15]. In plants, flavonoids are synthesized mainly via the phenylpropane metabolic pathway, which involves several key enzymes (phenylalanine ammonia-lyase (PAL), cinnamate-4-hydroxylase (C4H), and ligase for 4-coumarate CoA (4CL), among others) and a complex metabolic network [16]. Plant germination triggers the activation of enzymes in biochemical reactions that are involved in the synthesis and release of bioactive compounds [17]. Black beans treated with slight acid significantly increased flavonoid content after germination by enhancing the activity of the key enzymes of flavonoid metabolism (e.g., 4CL and C4H) and related gene expression [10]. In addition, flavonoids in plants can also increase the synthesis of certain metabolites by stimulating changes in gene expression and increasing the synthesis of certain metabolites. The synergistic effects of buckwheat during germination via caffeic acid, L-Phe, and NaCl promote the increase in flavonoid content mainly by promoting phenylpropane biosynthesis and the synthesis of various intermediates in the metabolic pathway of flavonoid biosynthesis [15]. The above studies illustrate that there are differences in flavonoid accumulation in different species, and the mechanism of flavonoid enrichment can be explored from a physiological and genomic perspective.

During the germination of seeds, germination conditions have a great influence on the synthesis of its secondary metabolites. Perales-Sánchez et al. [18] optimized the germination time and germination temperature of amaranth seeds to maximize their antioxidant activity, total phenol, and flavonoid content. Different types of barley at different germination temperatures and times showed different variations in total phenolic content after germination [19]. Dey et al. [20] optimized the ultrasonic treatment time, germination time, and temperature on the obtained kodo, and little millets had the highest antioxidant activity and lowest phytic acid content. Therefore, controlling the germination conditions of plants is highly effective for the accumulation of secondary metabolites in plants. Despite the various studies on the effect of germination on the nutritional value of small grains such as finger millet [12,13], there are no studies on optimizing the germination conditions of finger millet to promote the content of specific secondary metabolites in flavonoids. In addition, the mechanism of flavonoid enrichment in paniculata sprouts after germination is unclear.

Therefore, in this study, response surface methodology (RSM) was used to optimize the germination conditions of finger millet. The changes in physiological characteristics of finger millet during the germination process were investigated to explore their relationship with the changes in flavonoid content. The activities of antioxidant enzymes, flavonoid synthesis-related enzymes, and gene expression were analyzed to further investigate the response mechanism of flavonoid synthesis in germinated finger millet. This study provides a technological basis for the potential of finger millet as a raw material of functional ingredients in food.

## 2. Results

### 2.1. Optimum of Germination Conditions by RSM

Based on the results of the one-way test (Appendix A), germination time, germination temperature, and illumination duration were determined, and 17 rounds of a three-factor, three-level Box–Behnken framework were used to optimize the germination treatment parameters and generate second-order response surfaces (Appendix A).

The significance test of the quadratic polynomial mode and the results of the analysis of variance are displayed in Table 1. The fit of the prediction model was evaluated by calculating the coefficient R^2^, which was determined to be 0.9989. The validity of the model was demonstrated by the fact that its *F*-value was significant (*p* < 0.05), and the lack of fit *F*-value was not significant (*p* > 0.05). Table 1 shows that the overall accumulation of flavonoids is significantly affected (*p* < 0.01) by germination time, germination temperature, and illumination duration. Furthermore, it was found that the interaction variables between germination time and germination temperature (*p* < 0.01) and between germination time and illumination duration (*p* < 0.01) were significant for the accumulation of flavonoids in finger millet sprouts. Significant differences were observed between secondary variables, germination time, germination temperature, and illumination duration (*p* < 0.01). Equation (1) was obtained from an analysis of the experiment using multiple regression.
(1)$$Y\;=6.94\;+0.07X_{1\;}+0.64X_{2}+0.08X_{3\;}{+0.17X}_{1}X_{2}-0.05X_{1}X_{3}+0.02X_{2}X_{3}\;-\;0.09X_{1}^{2}\;-\;0.96X_{2}^{2}\;-\;0.14X_{3}^{2}$$
where Y is the predicted value of flavonoid content and X_1_, X_2_, and X_3_ represent the coded values of germination time, germination temperature, and illumination duration, respectively.

According to Figure 1 and Equation (1), 5.7 d of germination time, 31.2 °C of germination temperature, and 17.5 h of illumination duration are the optimal germination conditions for increasing the flavonoid content in the finger millet sprouts. This condition resulted in the highest flavonoid content of 7.0 μg/sprout.

### 2.2. Effects of Germination Time on Morphology, Flavonoid Content, Total Phenolic Content, and Anthocyanin Content in Finger Millet Sprouts

The morphology of the finger millet sprouts of different germination times is shown in Figure 2I. Compared with the seed, the flavonoid content in the 3 d and 5 d sprouts was 3.27 times and 5.18 times higher, respectively (Figure 2II). At 3 d, the total phenolic content and anthocyanin of the finger millet sprouts were significantly increased compared with those of ungerminated seeds (*p* < 0.05) (Figure 2III,IV). Germination promoted the accumulation of flavonoids, total phenolics, and anthocyanins in the finger millet. 

### 2.3. Effect of Germination Time on the Malondialdehyde (MDA) Content in Finger Mibbllet Sprouts

As shown in Figure 3, germination significantly increased MDA content in finger millet sprouts of 3 d (*p* < 0.05), which was 1.50 times that in seeds. With the prolongation of germination time, the content of MDA began to decrease on the 5th day, and there was no significant difference from that of seeds (*p* > 0.05). MDA content was used as an index to reflect the degree of membrane peroxidation in plants, which indicated that the germination stage was accompanied by the first strengthening of the membrane peroxidation degree and then some relief.

### 2.4. Effects of Germination Time on the Antioxidant Enzyme Activity and Antioxidant Capacity in Finger Millet Sprouts

As shown in Figure 4, The activities of antioxidant enzymes were all significantly enhanced (CAT, APX, POD, and SOD) in germinated finger millet compared to ungerminated seeds (*p* < 0.05) (Figure 4I–IV). Among them, CAT, APX, and POD activities were all highest at 3 d of germination time and were 5.97, 572, and 5.93 times higher than those of seeds, respectively. The DPPH radical scavenging capacity and FREP ferric reducing capacity were significantly higher in finger millet sprouts compared to the seeds (*p* < 0.05) (Figure 4V,VI). The above results indicated that sprouting enhanced the activity of antioxidant enzymes and improved the antioxidant capacity in finger millet sprouts.

### 2.5. Effects of Germination Time on Flavonoid Synthetase Activities in Finger Millet Sprouts

Changes in key enzymes for flavonoid synthesis during germination of finger millet sprouts are shown in Figure 5. The activities of PAL and 4CL in sprouts were the highest at 3 d, which were 1.18 and 1.30 times higher than those of the seeds (Figure 5I,II). The enzyme activity of C4H was the highest at 5 d, which increased by 92% compared with that of 0 d seeds (Figure 5III). All three enzymes reached higher activities after germination, although the changes were not consistent, which is consistent with the accumulation of flavonoids after germination.

### 2.6. Changes of Relative Expression Levels of Genes of Key Enzymes for Antioxidant Oxidase and Flavonoid Synthesis in Finger Millet Sprouts

Changes in the expression levels of genes related to the antioxidant system and phenylpropane metabolic pathway in finger millet sprouts during germination are shown in Figure 6. The expressions of *CAT* and *APX* were significantly up-regulated (*p* < 0.05) in 3 d germinated finger millet sprouts compared to the ungerminated seeds. Surprisingly, the expression of *APX* was increased 177-fold and 57-fold in 3 and 5 d germinated sprouts compared to seeds, respectively, while the expression of *SOD* was significantly down-regulated (*p* < 0.05).

The expression of *PAL*, *C4H*, and the transcription factor *MYB* was significantly up-regulated (*p* < 0.05) in finger millet sprouts after germination compared to seeds and was maximum at 3 d, with a 22-fold, 145-fold, and 8.93-fold increase, respectively. In addition, the relative gene expressions of three genes (*4CL*, *CHS*, and *CHR)* were also significantly up-regulated (*p* < 0.05) after germination compared to 0 d seeds. Their expressions reached a peak at 5 d of germination time, which was 33-fold, 27-fold, and 8.84-fold higher than that of the seeds, respectively. Only *CHI* was significantly down-regulated (*p* < 0.05) in relative gene expression in finger millet after germination.

### 2.7. Correlation between Flavonoid Content and Other Indexes of Finger Millet during Germination

The correlation analysis of flavonoid content with physiological and biochemical indexes and flavonoid synthesis-related indexes during the germination process of finger millet was carried out by calculating Pearson’s correlation coefficient, and the results are shown in Figure 7. Flavonoid content was positively correlated with the activities of SOD (*p* < 0.05). In addition, the activities of the C4H enzyme and *4CL*, *CHS*, and *CAT* genes were positively correlated with flavonoid content. The activity of the PAL enzyme and the expression of *CHI* genes were negatively correlated with flavonoid content (*p* < 0.05). Flavonoid content was also positively correlated with the antioxidant index DPPH’s radical scavenging capacity (*p* < 0.01).

## 3. Discussion

In plants, flavonoids play multiple protective roles, helping plants protect against adverse external factors such as ultraviolet radiation, oxidative stress, and temperature fluctuations [21]. In addition, flavonoids also play a key role in early plant development, such as the germination stage, participating in signaling processes and, if necessary, transforming into substances with defensive functions in response to biological stresses such as insects [22]. For humans, flavonoids act as a natural antioxidant and have significant health benefits [23]. Through a simple sprouting operation, we can easily obtain finger millet sprouts rich in flavonoids. The results (Figure 1) showed that with the increase in germination time, germination temperature, and illumination duration, the flavonoid content in finger millet sprouts increased first and then decreased. Under the optimal conditions obtained by the response surface methodology, the flavonoid content in finger millet sprouts was up to 7.0 μg/sprout.

The germination of seeds is accompanied by various biochemical reactions. In the initial stage of germination, the seeds absorb a lot of water, start to change from the dormant state, and gradually accelerate their metabolism. Plant respiration generates the energy for the reaction, accompanied by products such as superoxide (O_2_^−•^), hydrogen peroxide (H_2_O_2_), and hydroxyl free radical (•OH), collectively referred to as reactive oxygen species (ROS) [24]. ROS are toxic and, on the other hand, a key component of signal transmission in plant cells. When the concentration does not reach the toxic level, they are transmitted as a signal molecule, triggering a series of signal transduction processes to cope with stress [25,26]. For the detailed changes in ROS content after germination of finger millet, we will use the Pasternak and method [27] to determine the contents of hydrogen peroxide and superoxide anions in the sprouts in the future. In addition, fluorescence microscopy will be used to locate H_2_O_2_ and O_2_^−•^ in different tissues, such as root tips and the cotyledon, further clarifying the effect of ROS on the germination of finger millet. ROS produced in cells induces lipid peroxidation of unsaturated fatty acids in the membrane. MDA is the end production of lipid oxidation in cell membranes during plant growth. The higher the content of MDA, the greater the damage to plant membrane permeability. The results in Figure 3 reflect the changes in MDA content in finger millet after germination. The variation in the trend of MDA content increased first and then decreased, which was basically the same as the result of Huang et al. [10]. The results showed that finger millet self-regulated during germination and reduced the damage of membrane peroxidation to cells.

To maintain a stable environment in the cell during germination, seeds will synthesize some secondary metabolites to help resist external aggression [28]. In the first five days of germination, the flavonoid content and anthocyanin content of finger millet gradually increased (Figure 2III,IV). At the beginning of germination, phenols and flavonoids may act as free radical scavengers or antioxidants, while later, they may become part of the structure of the newly grown plant and lose some of their antioxidant efficiency. With the extension of germination time, sugar, crude fiber, ascorbic acid, and antioxidant activity increased significantly [5]. The antioxidant capacity of finger millet after germination was measured; the DPPH free radical clearance of finger millet sprout was 70.00–72.14%, and the FRAP reducing capacity was 53.69–53.76 mg/g [29]. The antioxidant capacity results in this study are consistent with the above studies (Figure 3V,VI).

After germination, the anti-nutrient factors were significantly decreased under different treatments in finger millet. The enhanced enzyme activity further led to the release and synthesis of bound phytochemicals (phenolic acids and flavonoids), thereby increasing antioxidant capacity several times [30]. In this study, the activities of antioxidant enzymes and key enzymes of flavonoid synthesis in finger millet were investigated during germination. The results (Figure 4I–IV) showed that the activities of four antioxidant enzymes (CAT, POD, SOD, and APX) were significantly increased compared with those of ungerminated seeds (*p* < 0.05). It was consistent with the conclusion that after germination, the antioxidant system of barley seedlings was enhanced, especially the activities of antioxidant enzymes [31]. Thus, the overall antioxidant performance of plants could be improved to resist the damage of external adverse factors.

The germination process effectively increased the antioxidant activity and total phenolic and flavonoid contents of finger millet seeds. This is consistent with the results in amaranth [18] and foxtail millets [11]. The increased antioxidant activity is due to the biosynthesis of polyphenols (phenolic acids and flavonoids) in the seeds. Singh et al. [32] noted that these bioprocessing treatments (soaking and germinating) were also characterized by the enzymatic degradation of the grain cell wall, resulting in the release of phenols and flavonoids. Cell wall-degrading enzymes are active during germination and help modify the cell wall structure of the grain. Phenolic compounds such as hydroxycinnamic acids (such as ferulic acid and p-coumaric acid) bind to non-starch polysaccharides in grain cell walls through ester and ether bonds [33,34]. The action of cell wall degrading enzymes on these bonds contributes to the release of bound phenolic compounds and flavonoids. In addition, it has been reported that polyphenol oxidase and peroxidase are activated at the beginning of germination, which may lead to the reduction in flavonoid compounds [35]. With prolonged germination, the increase in flavonoid compounds may be attributed to the activation of various flavonoid synthetases in germinated proso millet [36]. In this study, flavonoid content gradually increased with the increase in days, and no trend of decrease was observed. It is possible that the seeds (0 d) in our study were dry seeds, which did not undergo the process of water absorption and seed coat bursting, and the endogenous enzymes were not activated. We will incorporate the soaked seeds into the control in the next study.

Flavonoid biosynthesis is one of the important secondary metabolic pathways in plants. Related studies mainly focus on the metabolism of flavonoid substances downstream of the phenylpropane pathway, and its content has a direct effect on plant growth traits [37]. Using genomics technology to study the changes in secondary metabolites can reveal the molecular mechanism of relevant important metabolic pathways [15]. Therefore, based on the relative expression changes of key enzyme genes and stress-related genes in the flavonoid synthesis pathway of finger millet during germination, we hope to further understand the causes of flavonoid accumulation. The results (Figure 6) showed that the relative gene expressions of stress-related genes *CAT*, *APX*, and *MYB* were up-regulated during germination. In previous studies on the germination of grass seeds, *EcCAT1*, *EcSOD*, and *EcAPX1* were up-regulated [31]. Humic acid may participate in the activation of the mitogen-activated protein kinase (MAPK) signal cascade and regulate reactive oxygen species (ROS) homeostasis by up-regulating the expression of *EcSOD* and *EcCAT1* genes in finger millet. This might be something we need to analyze further in the next work. MYB family plays an important role in plant growth and development, stress resistance, and secondary metabolite synthesis [38]. For example, the *MYB12* of Arabidopsis regulated flavonoid biosynthesis by activating Chalcone synthetase (CHS) [39]. The gene expressions of *MYB* and *CHS* were up-regulated after germination (Figure 6), which may be attributed to the role of the MYB transcription factor in promoting the synthesis of flavonoids. Other key enzymes in flavonoid synthesis, including *PAL*, *C4H*, *4CL*, *CHS*, and *CHR*, were significantly up-regulated after germination (Figure 6) (*p* < 0.05). The above results were basically consistent with those of Yin et al. [40], except that *POD* expression decreased. The results of slight acid treatment showed that with the increase in germination, gene expression levels of most of the flavonoid synthetase in black beans were up-regulated, while only the relative gene expression levels of CHR and IFR were down-regulated [10]. Our study results (Figure 6) were slightly different but overall similar, and the changes in gene expression levels of most flavonoids and anabolic enzymes were up-regulated. The expression levels of *IFR* and *CHI* genes were down-regulated. This indicates that the synthesis of secondary metabolites and the regulation of some important genes in plants under germination have important effects.

Based on the above results and discussion, we speculated a putative mechanism of the stimulation of flavonoid biosynthesis by finger millet after germination (Figure 8). After germination, MDA content in finger millet sprouts increased first and then decreased. Germination activated the antioxidant system of finger millet sprouts, improved the activity of four antioxidant enzymes and their gene expression, and alleviated cell damage. The increase in PAL, C4H, and 4CL activities and the up-regulation of related genes were the main factors promoting the synthesis of flavonoids in millet after germination. Germination also increased the contents of secondary metabolites such as total phenolics and anthocyanins. These two secondary metabolites, together with flavonoids, promoted the antioxidant capacity of sprouts.

## 4. Materials and Methods

### 4.1. Plant Materials and Cultivation Conditions 

Finger millet seeds were sterilized and soaked. A portion of dried, impurity-free finger millet seeds (purchased in 2023 from Chayu County, Tibet Autonomous Region, China) were soaked in 0.5% sodium hypochlorite for 15 min, followed by rinsing with deionized water until reaching a neutral pH. Subsequently, the seeds were immersed in deionized water for 7 h. The soaked seeds were then placed in germination trays (340 mm × 250 mm × 120 mm) lined with wet gauze and then placed in a germination chamber (KM-68S, Kemai Instrument, Ningbo, China). The germination chamber can set the germination temperature and provide a fixed illumination intensity light source of 635 μmol/m^2^/s. The starting time was put into the germination chamber, and 30 mL of deionized water was sprayed every 12 h during germination. After germination, the fresh sprouts were put into polyethylene bags, treated with liquid nitrogen, and stored at −20 °C for further analysis. 

### 4.2. Optimization of Germination Conditions Using RSM

Following single-factor testing, the starting range of germination time, germination temperature, and illumination duration was established (data are displayed in Appendix A). Three dependent variables were included in the Box–Behnken study design. The independent variables were established as follows (with low and high values): germination temperature of 27–33 °C, germination time of 4–6 d, illumination duration of 8–24 h. Design-Expert 13 (State-Ease Inc., Minneapolis, MN, USA) was used to examine the experimental design and generate the expected data to estimate the response of the independent variables.

### 4.3. Extraction and Determination of Flavonoids

The flavonoid content was determined by the method of Vélez et al. [41]. A total of 1.0 g of fresh finger millet sprouts were ground with 80% (*v*/*v*) ethanol solution into a homogenate with no obvious plant tissue. Then, the homogenates were sonicated for 25 min at 25 °C. After centrifuging at 9000× *g* for 10 min, 0.5 mL of supernatant was taken and diluted 25 times with 80% (*v*/*v*) ethanol solution. The absorbance was measured at 260 nm, and genistein was used as an equivalent to make a standard curve.

### 4.4. Determination of the Content of Total Phenolic and Anthocyanins

Total phenolic content was measured according to the method of Wang et al. [42], with minor modifications. The sprouts (1.0 g) were extracted with 5 mL 50% (*v*/*v*) methanol and centrifuged at 10,000× *g* for 15 min. Supernatant (1 mL) was mixed with 1 mL 0.2 mM Folin–Ciocalteu reagent and 2 mL of 2% (*w*/*v*) Na_2_CO_3_ for 2 h in darkness. Total phenolic content was determined by reading the absorbance at 765 nm. Gallic acid was used as the standard.

The anthocyanin content was measured according to the method of Francis et al. [43] with a slight modification. The sprouts (0.5 g) were extracted with acidified ethanol (85:15 of 95% ethanol:1.5 M HCl) and centrifuged at 10,000× *g* for 15 min. Supernatant was collected, and its absorbance measured at 535 nm. Cyanidin-3-glucoside was used as the standard material. 

### 4.5. Determination of the Content of Malondialdehyde

The contents of malondialdehyde (MDA) were determined according to Yin et al. [40]. Initially, 1.0 g of finger millet sprouts were ground with the addition of 5.0 mL of 5% trichloroacetic acid. Subsequently, the resulting homogenate was subjected to centrifugation at 8000× *g* for 10 min. Following centrifugation, 2.0 mL of 0.76% thiobarbituric acid was introduced to the supernatant, which was fully mixed and boiled in a 100 °C water bath for 30 min. After cooling, the supernatant was separated, and its absorbance was measured at 450 nm, 532 nm, and 600 nm, respectively. 

### 4.6. Determination of Antioxidant Activity

The radical scavenging activity of 1,1-diphenyl-2-picrylhydrazyl (DPPH) was measured according to the method of Yang et al. [44]. Firstly, 1.0 g of fresh millet sprouts were extracted with 5 mL of 50% (*v*/*v*) methanol and then centrifuged at 3000× *g* for 30 min. DPPH (2 mL, 0.2 mM) in ethanol was added to 1 mL of the extraction solution. The absorbance was measured at 517 nm after 30 min of incubation at 25 °C. Distilled water was used as the control. The scavenging activity of DPPH radicals by the sample was calculated according to the following Equation (2):(2)$$\mathrm{DPPH}\;\mathrm{scavenging}\;\mathrm{activity}\;(\%)=\left[\left(1-\mathrm{absorbance}\;\mathrm{of}\;\mathrm{sample}\right)÷\mathrm{absorbance}\;\mathrm{of}\;\mathrm{control}\right]\times 100$$

The ferric-reducing antioxidant power (FRAP) was determined according to the method of Zhou et al. [45]. Briefly, 100 µL of the extract was placed in a test tube, and the volume was adjusted to 1 mL with methanol. Phosphate buffer (2.5 mL 0.2 M, pH 6.6) and 2.5 mL 1% potassium ferricyanide were added to the tube and vortexed. The mixture was left for 20 min at 50 °C in a water bath. After incubation, 2.5 mL 10% (*w*/*v*) trichloroacetic acid was added, and the mixture was centrifuged at 8000× *g* for 20 min. Then, 2.5 mL of the supernatant was taken and mixed with 2.5 mL distilled water and 0.5 mL 0.1% (*w*/*v*) ferric chloride in a test tube, and the absorbance was measured at 700 nm. A higher absorbance value indicates a higher reducing power.

### 4.7. Determination of Antioxidant Enzyme Activity and Flavonoid Synthetase Activity

The sprouts (0.5 g) for catalase (CAT), superoxide dismutase (SOD), and POD activities analysis were ground under ice bath conditions with 50 mmol/L sodium phosphate buffer (pH 7.0) containing 1% soluble polyvinyl pyrrolidine. The sprouts (0.5 g) for Ascorbate peroxidase (APX) activity analysis were homogenized in a solution containing phosphate buffers (0.1 mol/L, pH 7.5), 0.5 mmol/L ethylenediaminetetraacetic acid, and 2 mmol/L ascorbic acid. The homogenized solution was centrifuged at 10,000× *g* at 4 °C for 15 min to extract the supernatant. The activities of CAT, SOD, and POD were determined according to the method of Yin et al. [40]. Ascorbate peroxidase (APX) assay was carried out by the method of Nakano and Asada [46].

The sprouts (0.5 g) were ground with 5 mL of Tris-HCl buffer (0.1 mmol/L, pH 8.9) and centrifuged at 10,000× *g* for 25 min at 4 °C. Phenylalanine ammonia lyase (PAL), cinnamic acid 4-hydroxylase (C4H), and 4-coumarate coenzyme A ligase (4CL) activities were determined according to the method of Wang et al. [47].

### 4.8. Determination of Relative Gene Expression Levels

A plant RNA Extraction Kit (9769, Takara, Kusatsu, Japan) was used to extract total RNA from frozen finger millet sprouts. Firstly, the sample was ground into a non-granular powder with liquid nitrogen. Then, total RNA was reverse transcribed into cDNA using the Prime Script ™ RT Master Mix Kit (RR092A, Takara, Japan). The ABI 7500 sequence detection system (Applied Biosystems, Foster City, CA, USA) was applied to quantitative assays using the SYBRR premix EX-Taq ™ (RR820A, Takara, Japan). All kits were used according to the instructions. Appendix A lists the primers that were utilized in the present study. The cycle threshold (Ct) values were normalized using the values of *Actin*. All samples’ Ct values were calculated and analyzed using the formula 2^−ΔΔCt^.

### 4.9. Statistical Analysis

The data in this study came from three repeated trials, and the results were analyzed by variance and expressed as mean ± standard deviation. The data analysis involved conducting correlation analysis and Tukey’s multiple tests using SPSS version 25.0 (SPSS Inc., Chicago, IL, USA), with a significance level of *p* < 0.05 and *p* < 0.01.

## 5. Conclusions

In this study, the RSM was used to optimize the conditions for finger millet germination, and the optimum germination conditions were obtained as follows: germination time of 5.7 d, germination temperature of 31.2 °C, and illumination duration of 17.5 h. Furthermore, the changes in physiological characteristics, antioxidant system, flavonoid synthesis-related enzymes, and gene expression of finger millet were studied before and after germination. It was found that germination significantly increased the MDA content in finger millet sprouts. Compared with 0 d seeds, the activities of antioxidant enzymes and flavonoid synthetase in sprouts were enhanced, along with the relative expressions of related genes, which ultimately promoted the synthesis of flavonoids and the antioxidant capacity. This study provided the basis for the regulation of secondary metabolism and the development of active substances after the germination of finger millet.

## Figures and Tables

**Figure 1 plants-13-02191-f001:**
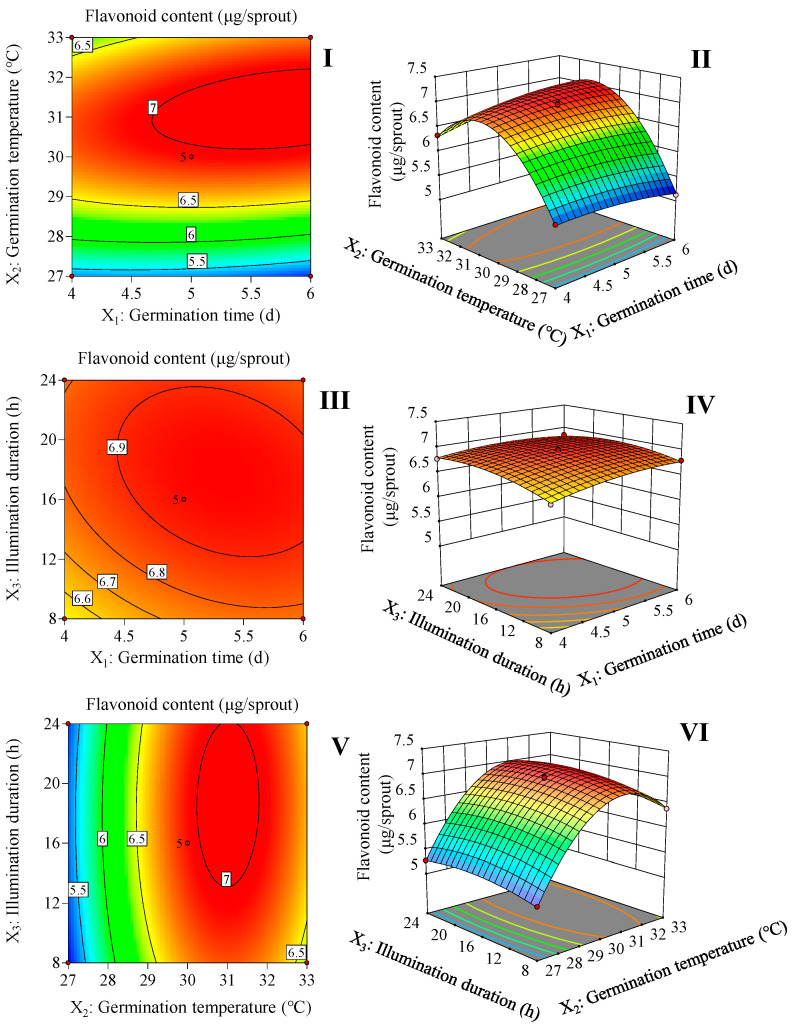
Response surface and contour map of the interaction of two factors on flavonoid content in finger millet sprouts. The interaction between the germination time and germination temperature (**I**,**II**). The interaction between the germination time and illumination duration (**III**,**IV**). The interaction between the germination temperature and illumination duration (**V**,**VI**).

**Figure 2 plants-13-02191-f002:**
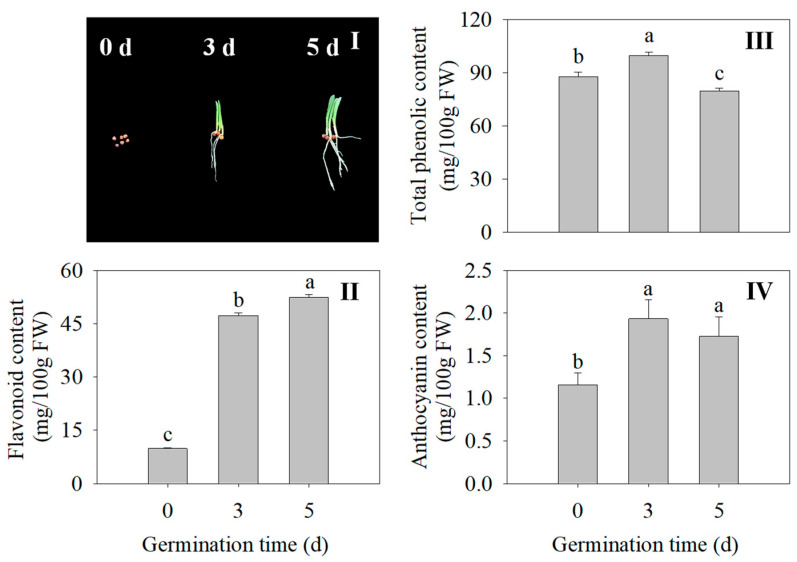
Effects of germination time on morphology (**I**), flavonoid content (**II**), total phenolic content (**III**), and anthocyanin content (**IV**) of finger millet sprouts. Different lowercase letters reflect the significance of differences in indexes between different germination times using Tukey’s test (*p* < 0.05). The germination time was set as 0, 3, and 5 d, the germination temperature was 31.2 °C, and the illumination time was 17.5 h. Subsequent tests were carried out under the same germinating conditions.

**Figure 3 plants-13-02191-f003:**
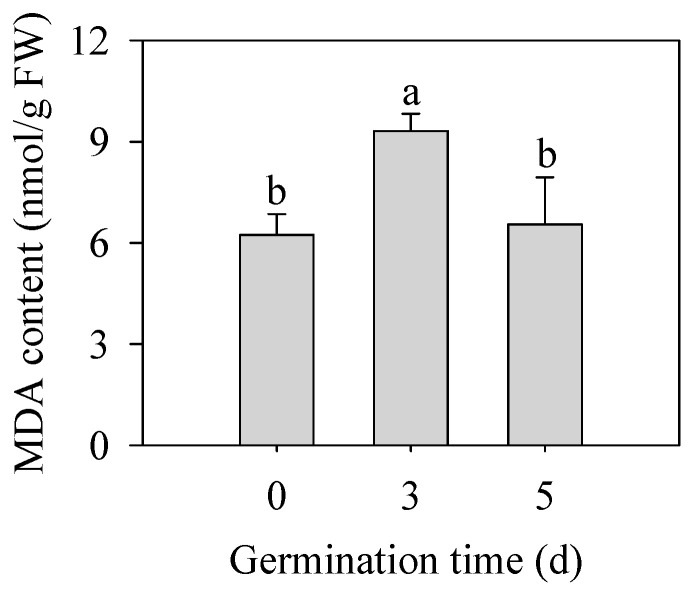
Effect of germination time on the contents of MDA in finger millet sprouts. Different lowercase letters reflect the significance of differences in indexes between different germination times using Tukey’s test (*p* < 0.05).

**Figure 4 plants-13-02191-f004:**
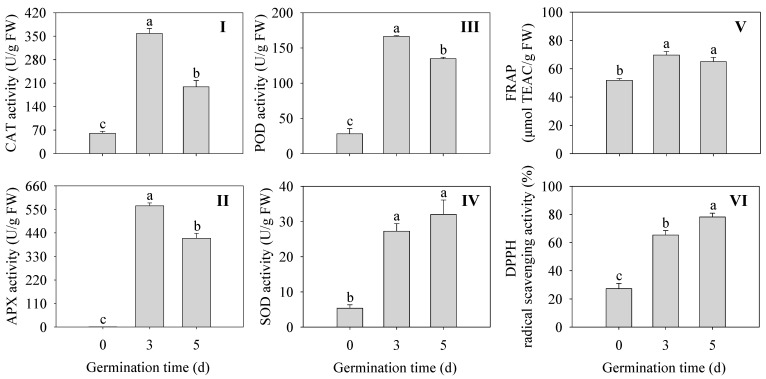
Effects of germination time on the antioxidant enzyme activity CAT (**I**), POD (**II**), SOD (**III**), and APX (**IV**), and antioxidant capacity PAL (**V**) and C4H (**VI**) in finger millet sprouts. Different lowercase letters reflect the significance of differences in indexes between different germination times using Tukey’s test (*p* < 0.05).

**Figure 5 plants-13-02191-f005:**
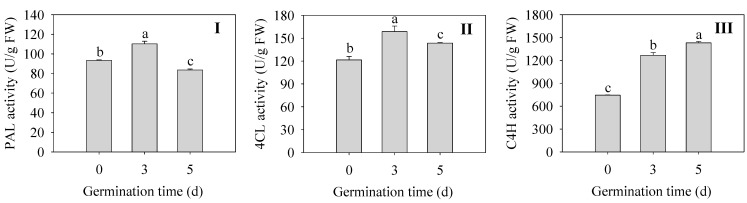
Effect of germination time on the activity of PAL (**I**), 4CL (**II**), and C4H (**III**) in finger millet sprouts. Different lowercase letters reflect the significance of differences in indexes between different germination times using Tukey’s test (*p* < 0.05).

**Figure 6 plants-13-02191-f006:**
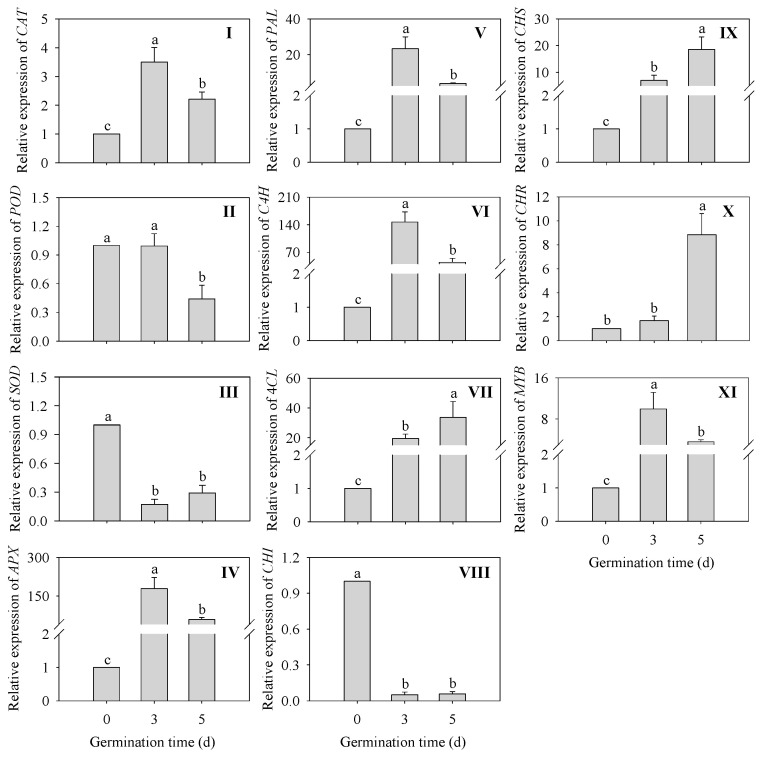
Effects of germination time in relative gene expression of *CAT* (**I**), *POD* (**II**), *SOD* (**III**), *APX* (**IV**), *PAL* (**V**), *C4H* (**VI**), *4CL* (**VII**), *CHI* (**VIII**), *CHS* (I**X**), *CHR* (**X**), and *MYB* (**XI**) in finger millet sprouts during germination. Different lowercase letters reflect the significance of differences in indexes between different germination times using Tukey’s test (*p* < 0.05).

**Figure 7 plants-13-02191-f007:**
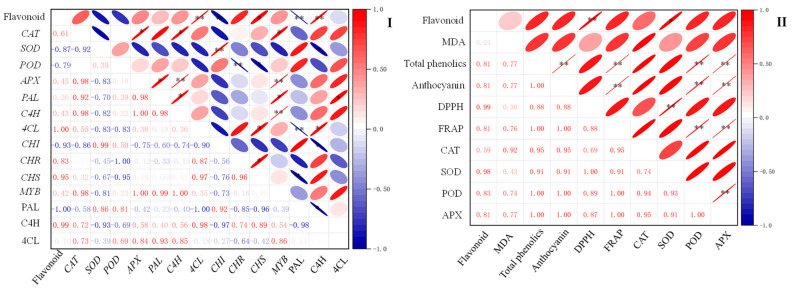
Correlation analysis of flavonoid content with flavonoid synthesis-related indexes (**I**) and physiological and biochemical indexes (**II**) during the germination process of finger millet. The negative and positive correlations are shown in different colors (blue and red). The correlation coefficients range from −1 to 1. The symbols * and ** are used to denote statistical significance at the *p*-values of 0.05 and 0.01, respectively, for the correlation coefficients.

**Figure 8 plants-13-02191-f008:**
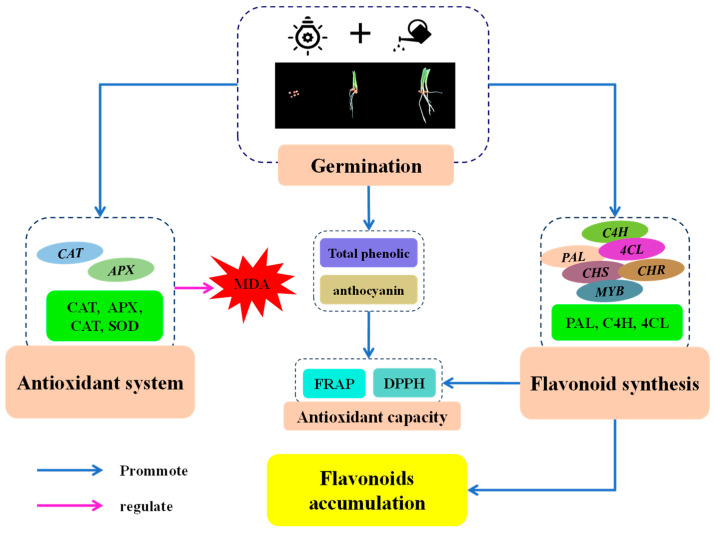
Putative mechanism of stimulation of flavonoid biosynthesis by finger millet after germination.

**Table 1 plants-13-02191-t001:** Test of significance for regression coefficient and analysis of regression model variance equation.

Source	Sum of Squares	df	Mean Square	*F*-Value	*p*-Value
Model	7.62	9	0.8466	702.31	<0.0001 **
X_1_: Germination time	0.0388	1	0.0388	32.19	0.0008 **
X_2_: Germination temperature	3.28	1	3.28	2723.19	<0.0001 **
X_3_: Illumination duration	0.0525	1	0.0525	43.52	0.0003 **
X_1_ X_2_	0.1130	1	0.1130	93.77	<0.0001 **
X_1_ X_3_	0.0122	1	0.0122	10.11	0.0155 *
X_2_ X_3_	0.0025	1	0.0025	2.07	0.1931
X_1_^2^	0.0321	1	0.0321	26.64	0.0013 **
X_2_^2^	3.88	1	3.88	3217.86	<0.0001 **
X_3_^2^	0.0771	1	0.0771	63.93	<0.0001 **
Residual	0.0084	7	0.0012		
Lack of Fit	0.0051	3	0.0017	2.00	0.2566
Pure Error	0.0034	4	0.0008		
Cor Total	7.63	16			
C.V. % = 0.5436 R^2^ = 0.9989		Adjusted R^2^ = 0.9975		Predicted R^2^ = 0.9887	

where * indicates a significant level (*p* < 0.05), ** indicates a highly significant level (*p* < 0.01).

## Data Availability

The data presented in this study are available on request from the corresponding author.

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
