# Peer review of "Germination Promotes Flavonoid Accumulation of Finger Millet (Eleusine coracana L.): Response Surface Optimization and Investigation of Accumulation Mechanism"

_plants, 2024, doi:10.3390/plants13162191_

Round 1
Reviewer 1 Report
Comments and Suggestions for Authors
In the manuscript submitted for review, the authors used the germination process to increase the level of polyphenolic compound synthesis in finger millet. The entire manuscript is correct and written in accordance with the adopted guidelines of the publisher. The abstract is correct and is a synthetic summary of the work. The introduction provides a good background describing the selected plant material and the germination process. The aim of the work has been clearly specified. The methodological section describes the methods used, which were appropriately selected for the planned experiment. I only have a comment on the section concerning the extraction protocol. The authors should supplement the information on the amount of plant material used for extraction. What were the extracts diluted with? What equivalents were used in the determination of flavonoids and athocyanins? There is also no technical data for the centrifuge.
I am also wondering about the results obtained regarding the content of individual classes of polyphenolic compounds. How is it possible that the authors showed an increase in flavonoid accumulation after 3 and 5 days of germination with a decrease in total polyphenols and anthocyanins on day 5? The authors should try to explain this mechanism. No precise conclusion can be drawn from the analysis of enzyme activity.
Author Response
Response to Reviewer
Comments 1
In the manuscript submitted for review, the authors used the germination process to increase the level of polyphenolic compound synthesis in finger millet.
Response:
Respected reviewer, thank you very much for your comments.
Comments 2
The entire manuscript is correct and written in accordance with the adopted guidelines of the publisher. The abstract is correct and is a synthetic summary of the work. The introduction provides a good background describing the selected plant material and the germination process. The aim of the work has been clearly specified. The methodological section describes the methods used, which were appropriately selected for the planned experiment.
Response:
Thank you for spending time in reviewing our manuscript and providing us with a list of constructive comments.
Comments 3
I only have a comment on the section concerning the extraction protocol. The authors should supplement the information on the amount of plant material used for extraction.
Response:
Thank you for pointing out this issue, which makes our extraction method more complete. We have modified the corresponding content in the original text to supplement the quality of the plant material in each extraction protocol. These revisions can be found on page 12 in lines 354-355, 362, 368, 375, 383, 399, and 401, on page 13 in lines 408.
Comments 4
What were the extracts diluted with?
Response:
Thank you for your scrutiny. We think what you want to know is the diluent used before the flavonoid assay. The dilution here used is 80% (v/v) ethanol solution, which we have added to the modified version on page 12 in lines 358.
Comments 5
What equivalents were used in the determination of flavonoids and anthocyanins?
Response:
Thank you for your insightful questions. According to your comments, we have made additions in the revision, which can be found in 359 and 359 lines, respectively. We used genistein as the equivalent when determining flavonoid content and Cyanidin-3-glucoside as the equivalent when determining anthocyanins. These revisions can be found on page 12 in lines 354-355, 362, 368, 375, 383, 399, and 401, on page 13 in lines 408.
Comments 6
There is also no technical data for the centrifuge.
Response:
We appreciate your valuable feedback. In the revised manuscript, we have expanded all the details involved in centrifugal operation, mainly centrifugal force (g), centrifugal time (min) and centrifugal temperature (℃), in which the temperature of centrifugation is only set at 4℃ when measuring enzyme activity, and the rest are room temperature.
Comments 7
I am also wondering about the results obtained regarding the content of individual classes of polyphenolic compounds. How is it possible that the authors showed an increase in flavonoid accumulation after 3 and 5 days of germination with a decrease in total polyphenols and anthocyanins on day 5? The authors should try to explain this mechanism. No precise conclusion can be drawn from the analysis of enzyme activity.
Response:
Respected reviewer, thank you very much for your comments.
The synthesis of the three polyphenols mentioned by the reviewers all requires the phenylpropane pathway involving PAL, C4H and 4CL. As our results show (Figure 5I-III and Figure 6V-VII), after germination, the activity of the three key enzymes and the corresponding gene expression are all improved, which verifies that the content of these three substances in finger millet sprouts is increased after germination. Polyphenols are a general term of a large category, and the total phenolic measured by us is also one of them. However, the changes of such polyphenols after germination are regulated by many external factors. Ren and Sun [1] observed that total phenolic content (TPC) and total flavonoid content (TFC) reached the highest point at different germination times in buckwheat during germination, which was like our study and may be the reason for the reviewers' questions. In addition, the difference of extraction solution can also cause the difference of total phenol content between samples. Islam et al. extracted total phenols from lupine beans with 80% ethanol and water respectively to obtain different contents of the same sample, and extracted more total phenols with 80% ethanol. In the process of the determination of total phenol in millet seeds and seedlings, 50% methanol was used in the extraction solution, which may also be the reason for the unsatisfactory results. The total phenolic content of these plants changes inconsistent during germination, so the results mentioned by the reviewers in our study can be acceptable. This kind of active substance plays an important role in plant growth regulation.
Anthocyanins belong to a kind of flavonoid, and the key enzymes in their synthesis pathway include phenylpropane pathway, as well as two genes "CHI, CHR, and CHS" in our results (Figure 6VIII-X), and the gene expression changes of these three enzymes are inconsistent.
In addition, we are more concerned about the isoflavone part of the flavonoid content changes, which can be seen from the standard we use (genistein). The content of various substances in plants is mainly regulated by the two pathways of synthesis and decomposition. IFS and IFR are isoflavone synthase and isoflavone reductase respectively. We did not show these two graphs (At the end of the paragraph) when we finished the manuscript. From these two diagrams, we can see that the relative gene expression levels of these two enzymes are inconsistent. The gene expression level of IFS is up-regulated at 3 d, while the gene expression level of IFR is down-regulated at 3 d, which is consistent with the flavonoid content in this study that shows an increase in flavonoid accumulation 3 and 5 days after germination (Figure 2II). Secondly, there was no significant change in the content of anthocyanin between 3 d sprouts with 5 d sprouts, and it could not be determined that the content of anthocyanin was reduced on the 5th day. Finally, I would like to point out that the germination time of 5 d will also reach the optimal time point for flavonoid enrichment. Therefore, the variation among the three phenolic substances is reasonable and can be clarified through our experimental results and reference to relevant literature. Your comments make our research ideas broader. We are grateful for your thoughtful comment.
Ren, S.C.; Sun, J.T. Changes in phenolic content, phenylalanine ammonia-lyase (PAL) activity, and antioxidant capacity of two buckwheat sprouts in relation to germination. J. Funct. Foods, 2014. 7, 298-304.
Islam, M.Z.; Kang, S.W.; Koo, N.G.; Kim, Y.J.; Kim, J.K.; Lee, Y.T.; Changes in Antioxidant Bioactive Compounds of Cassia Tora Linn. Seed During Germination. Cogent Food Agr. 2023. 9, 220227.

Reviewer 2 Report
Comments and Suggestions for Authors
The authors performed hard work to study germination process in finger Millet. The authors focused on flavonoid accumulation and made an attempt to describe the mechanism. Modern response surface optimization method has been used.
Despite of many interesting data, the paper require significant revisión and re-evaluation.
The key unclear point is what did authors mean as “0” point.
There is no sprout at this time. And one can not compare dry seeds (or seeds after 7 h priming¿???) with sprouts.
Moreover, the normalization is also nuclear. On fig 2 it is mkg/sprout, or nmol/g FW and nmol/100 g FW.
Figs 3 and 4 are per g of FW, what is complettely wrong, mainly if you significantly change wáter contents from 0 to day 3. 0 time points is overestimate on figure 3, but can underestimate on figure 4.
On figure 6 authors used some relative units (per what)¿?
Correspondingly, discusión and conclusión need to be re-evaluated.
Some details:
There is a mistake on line 16: it is not clear what did authors mena as 17.5 h. Later on I have found that it is illumination duration, but this should be mentioned in the abstracts as well.
Line 26: “the seeds were treated with germination” ¿?? Germination is not a treatment!
Line 326: why after title 4 come subtitle 2.1?
Line 334: 30 ml per what? What is germination tray size?
Line 344: Illumination intensity?
Line 349: 1 g or 100 mg were ground?
Line 372: wáter bath temperature??
Lines 380 – 386: this is the most importnat Discovery of the current paper. Base don M&M reader can conclude that superoxide is stable at least 2 hours and half-life is veren longer. This is very new properties of ROS and should be proven independently.
Plesae, provide direct evidences of this new phenomena. It is not necessary to provide more citation of others who claim similar superoxide chemistry, in this citation there are no any evidence how did authors of cited paper extend superoxide stability. Only authors own data are required.
Line 404: “flaonid” ¿?
Line 409: for APX one need to add Ascorbic acid in homogeniozation buffer.
Comments on the Quality of English Languagesome typos need to be corrected
Author Response
Kind reviewer.
Thank you for your valuable comments and suggestions on our manuscript (ID: plants- 3104360) ' Germination promotes flavonoid accumulation of finger millet (Eleusine coracana L.): Response surface optimization and investigation of accumulation mechanism '. We have carefully revised the manuscript and resubmitted it to you for further review. (All changes can be seen in the latest manuscripts).
Response to Reviewer
Comments 1
The authors performed hard work to study germination process in finger Millet. The authors focused on flavonoid accumulation and made an attempt to describe the mechanism. Modern response surface optimization method has been used.
Response:
Thank you for your positive evaluation and for your comments and suggestions that have helped to substantially improve our study.
Comments 2
Despite of many interesting data, the paper require significant revisión and re-evaluation.
Response:
Thank you for spending time in reviewing our manuscript and providing us with a list of constructive comments.
Comments 3
The key unclear point is what did authors mean as “0” point. There is no sprout at this time. And one can not compare dry seeds (or seeds after 7 h priming?) with sprouts.
Response:
We gratefully acknowledge your insightful comments. The authors regret that the specific meaning of 0 d seed was not explained in the manuscript. As the reviewers think, the germinated 0 d seed used in this study is equivalent to the ungerminated dry seed (stored in the refrigerator at -20℃ before use). The selection of this sample as the comparison of 0 d is considered as follows:
- We fully understand the reviewers' concerns, and we also know that a series of changes in millet after germination are the focus of this study, so we have done a lot of work on physiology, biochemistry and gene expression. In this study, 0 d seeds serve as the state of samples before germination, which is a very stable control (especially the dry seeds we selected). This study aims to explore the accumulation mechanism of flavonoids in finger millet sprouts. Comparison before and after germination can help us better understand the mechanism, and the results we obtained also indicate that 0 d seeds are a good choice for control.
- Many studies had selected 0 d(h) seeds as research samples to help them carry out research more scientifically. For example, Udeh et al. [1] studied the changes of phenolic substances and antioxidant activity of millet sprouting 0-96 h. Lan et al. [2] studied the effects of different germination stages (0-72 h) on the morphological characteristics, approximate composition, amino acid profile, GABA level, antioxidant properties, polyphenol content, and volatile compounds of quinoa. Garnczarska et al. [3] explored the relationship between antioxidant enzymes and ROS in dried seeds and germinated lupine seeds from a physiological and biochemical perspective. In addition, our previous studies [4] also investigated the enrichment mechanism of isoflavones in black beans during 0-24 h of germination time.
Therefore, it is reasonable and scientific to select 0 d seeds as a control for physiological and biochemical properties of finger millet during germination.
[1] Udeh, H.O.; Duodu, K.G.; Jideani, A.I. Malting Period Effect on the Phenolic Composition and Antioxidant Activity of Finger Millet (Eleusine coracana L. Gaertn) Flour. Molecules, 2018. 23, 2091.
[2] Lan, Y.; Xinze W.; Lei, W.; Wengang, Z., Yujie, S.; Shiyang, Z.; Xijuan, Y.; Xuebo, L. Change of Physiochemical Characteristics, Nutritional Quality, and Volatile Compounds of Chenopodium Quinoa Willd. During Germination. Food Chem. 2024. 445, 138693.
[3] Garnczarska, M.; Wojtyla, L. Differential response of antioxidative enzymes in embryonic axes and cotyledons of germinating lupine seeds. Acta Physiol. Plant, 2008. 30, 427-432.
[4] Huang, C.M.; Quan,X.L.; Yin, Y.Q.; Ding, X.L.; Yang, Z.F.; Zhu, J.Y.; Fang,W.M. Enrichment of Flavonoids in Short-Germinated Black Soybeans (Glycine max L.) Induced by Slight Acid Treatment. FOODS, 2024. 13, 868.
Comments 4
Moreover, the normalization is also nuclear. On fig 2 it is mkg/sprout, or nmol/g FW and nmol/100 g FW.
Response:
Thank you for making us notice. We have unified the units in Figure 2 to “mg/100 g FW” based on the suggestions of reviewers and referring to the result (Fig. 5) of Chen et al. [5] Changes to Figure 2 can be found on page 5 of the revised manuscript.
[5] Che, G. H.; T. Jiang, X. D.; Li, J. X.; Xiao, L.; Liu, J. T.; Wei, L.; Guo, P. Effect of Plasma Activated Water Immersion on Broccoli Seed Germination and Nutritional Quality of Sprouts. J. PLANT GROWTH REGUL. 2024. 43, 2373-2382.
Comments 5
Figs 3 and 4 are per g of FW, what is complettely wrong, mainly if you significantly change wáter contents from 0 to day 3. 0 time points is overestimate on figure 3, but can underestimate on figure 4.
Response:
Thank you for your constructive comments. We agree with you that the use of fresh weight as a unit for some indicators in the manuscript is indeed inadequate compared to the use of protein or dry weight as a unit. According to your suggestion, we will use dry weight or vigor in future studies to make the results more accurate. However, we did not measure the dry weight or protein content in this manuscript, so it is impossible to convert the units again. We have strengthened the discussion on this in the discussion section of the revised manuscript. We also noticed that some relevant studies [6-10] also used fresh weight as a unit to determine the relevant indicators.
Thank you again for pointing out your valuable questions.
[6] Chun, X.L.; FENG, S.L.; Yun, S.; JIANG, L.N.; LU, X.Y; HOU, X.L. Effects of arsenic on seed germination and physiological activities of wheat seedlings. J. Environ. Sci. 2007. 19, 725–732. doi:10.1016/s1001-0742(07)60121-1
[7] Salah, H.A.; Elsayed, A.M.; Bassuiny, R.I.; Abdel-Aty, A.M.; Mohamed, S.A. Improvement of Phenolic Profile and Biological Activities of Wild Mustard Sprouts. Sci. Rep. 2024.14,
10528. https://doi.org/10.1038/s41598-024-60452-5
[8] Elsherif, D.E.; Safhi, F.A.; Subudhi, P.K.; Shaban, A.S.; El-Esawy, M.A.; Khalifa, A.M. Phytochemical Profiling and Bioactive Potential of Grape Seed Extract in Enhancing Salinity Tolerance of Vicia faba. Plants 2024, 13, 1596. https://doi.org/10.3390/plants13121596
[9] Wang, X.F.; Zhao, J.Q.; Yuan, P.G.; Ding, S.Y.; Jiang, L.G; Xi, Z.M. Hydrogen Peroxide Functioned as a Redox Signaling Molecule in the Putrescine-Promoted Drought Tolerance in Cabernet Sauvignon. Sci. Hortic. 2024. 335, 113325. https://doi.org/10.1016/j.scienta.2024.113325
[10] Elbouzidi, A.; Taibi, M.; Baraich, A.; Haddou, M.; Loukili, E.H.; Asehraou, A.; Mesnard, F.; Addi, M. Enhancing Secondary Metabolite Production inPelargonium graveolens Hort. Cell Cultures: Eliciting Effects of Chitosan and Jasmonic Acid on Bioactive Compound Production. Horticulturae 2024, 10, 521. https://doi.org/10.3390/horticulturae10050521
Comments 6
On figure 6 authors used some relative units (per what)?
Response:
We appreciate your valuable feedback. Based on the relative unit of gene expression, the value of the relative expression of each sample was equal to the mean expression of the target gene minus the mean expression of the internal reference gene (Actin). Comparative Ct value method (2-ΔΔCt) was used to perform the relative quantitative calculation and statistical analysis of gene expression. We have made corresponding additions in line 419-420 of the revised manuscript.
Comments 7
Correspondingly, discusión and conclusión need to be re-evaluated.
Response:
Thank you for pointing out this issue. We have substantially revised two paragraphs and clarified the text.
- For the discussion part, we have made the following modifications:
- Considering the applicability of the method for measuring ROS (H2O2 and O2‾•) content (H2O2 and O2‾•) and the large discrepancy between our results (Original manuscript, Figure 3II and III, H2O2 and O2‾• content) and the relevant research results, we decided to delete the relevant results after careful consideration. The new description of oxidative stress caused by germination mainly explains ROS, and three new references (References 25, 26, and 27) are added. Two (References 25 and 26) on the sources and effects of ROS, one (References 27) on the methods of ROS determination, and the shortcomings and future directions of this study are summarized and prospected. In addition, ROS was also used to guide the formation of MDA, furthermore analyzing the results (Revised manuscript,Figure 3 ) of this study. Finally, it was concluded that during the germination of finger millet, cells were subjected to potential oxidative stress, which was manifested by the content of MDA. Based on its change trend, it was concluded that cell damage was regulated by plants themselves. Above modification can be found on page 9, line 243-263 of the revised manuscript
- Figure 8 has been modified to delete the conclusion on regulating ROS. The description of the enrichment mechanism speculated in Figure 8 has been rewritten, with ROS related discussion deleted and the statement expressed more briefly. These revisions can be found on page 11, line 317-326 of the revised manuscript
- As for the conclusion, we have revised it, and the details are as follows:
To avoid too much overlap with the abstract, we have deleted some superfluous comments, as well as the ROS metrics. The findings of this study are summarized again. For detailed revisions, please refer to page 13, line 429-437 of the revised manuscript
Comments 8
Some details: There is a mistake on line 16: it is not clear what did authors mena as 17.5 h. Later on I have found that it is illumination duration, but this should be mentioned in the abstracts as well, as seen in page
Response:
Thank you for your careful comments and suggestions. We supplemented the specific names of germination parameters in the revised manuscript, which can be found in page1, line 15-16.
Comments 9
Line 26: “the seeds were treated with germination”? Germination is not a treatment!
Response:
Thank you for pointing out the obvious error in our presentation of manuscript. We have removed this error and rewritten some of the content in the abstract in the revised manuscript, as seen in page1, line 22-26.
Comments 10
Line 326: why after title 4 come subtitle 2.1?
Response:
The authors are sorry for our careless mistakes. These mistakes have been corrected in revised manuscript, as seen in line 331, 344, 353, 360, 372, 381, 398, 412, and 422.
Comments 11
Line 334: 30 ml per what? What is germination tray size?
Response:
Thank you for your careful comments. We sprayed 30 mL of deionized water (from laboratory pure water preparation machine) every 12 hours during germination. The size of the germination tray we used is 340 mm×250 mm×120 mm. We have revised it with reference to your comments, which can be seen in page 11, line 336-337, 340-341.
Comments 12
Line 344: Illumination intensity?
Response:
Thank you for your serious comments. The illumination intensity we used is 635 μmol/m2/s. This key parameter has been added to the revised manuscript in page 11, line 339-340.
Comments 13
Line 349: 1 g or 100 mg were ground?
Response:
The quality of the plant material used in the flavonoid extraction process is about 1.0 g. This parameter can be found in page 12, line 354-355 of the revised manuscript. Thank you for your careful comment.
Comments 14
Line 372: wáter bath temperature??
Response:
In the determination of malondialdehyde, the temperature of the mixed liquid water bath is 100°C. This revision can be found on line 377 of page 12 of the revised manuscript. Thank you for your careful review which makes our method more complete.
Comments 15
Lines 380 – 386: this is the most importnat Discovery of the current paper. Base don M&M reader can conclude that superoxide is stable at least 2 hours and half-life is veren longer. This is very new properties of ROS and should be proven independently.
Plesae, provide direct evidences of this new phenomena. It is not necessary to provide more citation of others who claim similar superoxide chemistry, in this citation there are no any evidence how did authors of cited paper extend superoxide stability. Only authors own data are required.
Response:
Thank you for your careful review and valuable comments.
The measurement of this index in this study was based on the description of Zhao et al. [11] and Elstner et al. [12], which determined the generation of O2‾• by monitoring the nitrate formed by hydroxylamine in the presence of O2‾• generator. Considering the methods (A scientific and comprehensive method for the determination and localization of ROS content in plants) used by Pasternak and Perez-Perez [13] to measure ROS and the comments received on recent papers submitted by our research group, we have removed this indicator from our manuscript. Nevertheless, this does not affect our relevant conclusions. It can also be seen from the changes of MDA content during millet germination that the sprouts suffered cell damage, which was reflected in the deepening of membrane peroxidation. With the delay of germination, this harm is mitigated. In the future, we will explore more ROS-related indicators by using the methods of Pasternak and Perez-Perez [9].
[11] Zhao, K.; Zhao, C.; Yang, M.; Yin, D. ZnCl2 treatment improves nutrient quality and Zn accumulation in peanut seeds and sprouts. Sci. Rep. 2020. 10, 2364.
[12] Elstner, E.F.; Heupel, A. Inhibition of nitrite formation from hydroxylammoniumchloride: A simple assay for superoxide dismutase. ANAL. BIOCHEM. 1976. 70, 616-620.
[13] Pasternak, T.K.; Perez-Perez, J. Optimization of ROS Measurement and Localization in Plant Tissues: Challenges and Solutions; protocols.io: Berkeley, CA, USA, 2021.
Comments 16
Line 404: “flaonid” ?
Response:
The authors are very sorry for the error due to our carelessness, we have changed “flaonid” into “flavonoid” in line 397 of the revised manuscript.
Comments 17
Line 409: for APX one need to add Ascorbic acid in homogeniozation buffer.
Response:
Thanks for your valuable comment. We have described in detail the grinding steps in the determination method of APX and add the related methods. These revisions can be seen in page 12, line 400-406.
Comments 18
some typos need to be corrected
Response:
Thank you for your comments. We have carefully checked the text and corrected the typos

Round 2
Reviewer 1 Report
Comments and Suggestions for Authors
The authors have taken into account all my comments and have also dispelled my previous doubts. I recommend the manuscript for publication
Author Response
Kind reviewer.
Thank you for your valuable comments and suggestions on our manuscript (ID: plants-3104360) ' Germination promotes flavonoid accumulation of finger millet (Eleusine coracana L.): Response surface optimization and investigation of accumulation mechanism '. We have carefully revised the manuscript and resubmitted it to you for further review. (All changes can be seen in the latest manuscripts).
Response to Reviewer
Comments 1
The authors have taken into account all my comments and have also dispelled my previous doubts. I recommend the manuscript for publication
Response:
Respected reviewer, thank you for your positive evaluation and for your comments and suggestions that have helped to substantially improve our study.

Reviewer 2 Report
Comments and Suggestions for Authors
Thank you for the very detailed answers and literature collections.
The paper is much solid now, but authors still need to consider some
more points.
About day 0 and use FW at day “0”.
I would suggest next time use as “0” seeds after first physical stage of germination – water uptake. This will adequately reflected changes from seeds to seedlings. And also consider that most relevant changes happens during first few hours (chromatin remodeling), before endosperm rupture. Once endosperm rupture happens, seedlings establish auxin gradients, cell fate were re-established and seeds just follow similar rules as plants growth (with small exception).
About FW normalisation: in the cited paper I have not seen any comparison FW of dry seeds with FW of 3 days seedlings. The best normalization method of course is per cell (per DNA) but even this metod have some precaution since cell in the organs is not equal and have a very different epigenetic and correspondingly, gene expression/protein activity. This need to be add to discussion in the current paper as well: flavonoids related with certain cell type and always one can ask to which oprocess related higher flavonoids: cell number or contents per cell. In situ data are required for the relevant conclusion.
Please, modify discussion correspondingly.
Citation 27: dx.doi.org/10.17504/protocols.io.bx49pqz6 Please, use correct citation.
Comments on the Quality of English LanguageMinor editing during proof-reading.
Author Response
Kind reviewer.
Thank you for your valuable comments and suggestions on our manuscript (ID: plants- 3104360) ' Germination promotes flavonoid accumulation of finger millet (Eleusine coracana L.): Response surface optimization and investigation of accumulation mechanism '. We have carefully revised the manuscript and resubmitted it to you for further review. (All changes can be seen in the latest manuscripts).
Response to Reviewer
Comments 1
Thank you for the very detailed answers and literature collections. The paper is much solid now, but authors still need to consider some more points.
Response:
Thank you for your positive evaluation and for your comments and suggestions that have helped to substantially improve our study.
Comments 2
About day 0 and use FW at day “0”.
I would suggest next time use as “0” seeds after first physical stage of germination –water uptake. This will adequately reflected changes from seeds to seedlings. And also consider that most relevant changes happens during first few hours (chromatin remodeling), before endosperm rupture. Once endosperm rupture happens, seedlings establish auxin gradients, cell fate were re-established and seeds just follow similar rules as plants growth (with small exception).
Response:
Thank you for spending time in reviewing our manuscript and providing us these constructive comments. We agree with the reviewer’s suggestions, and we will adopt your suggestion in the following work.
Comments 3
About FW normalisation: in the cited paper I have not seen any comparison FW of dry seeds with FW of 3 days seedlings. The best normalization method of course is per cell (per DNA) but even this metod have some precaution since cell in the organs is not equal and have a very different epigenetic and correspondingly, gene expression/protein activity. This need to be add to discussion in the current paper as well: flavonoids related with certain cell type and always one can ask to which oprocess related higher flavonoids: cell number or contents per cell. In situ data are required for the relevant conclusion. Please, modify discussion correspondingly.
Response:
We gratefully acknowledge your insightful comments. In the discussion part of the revised manuscript, we have strengthened the discussion on the reasons why flavonoids are promoted during germination, and cited the corresponding literature to demonstrate. The revision is on page 10, line 288-307.
Comments 4
Citation 27: dx.doi.org/10.17504/protocols.io.bx49pqz6 Please, use correct citation.
Response:
Thank you for pointing out this issue.
We found an example [1] that cited this reference and revised it in our revised manuscript.
[1] Tian, X.; Zhang, R.; Yang, Z.; Fang, W. Methyl Jasmonate and Zinc Sulfate Induce Secondary Metabolism and Phenolic Acid Biosynthesis in Barley Seedlings. Plants 2024, 13, 1512. https://doi.org/10.3390/plants13111512
Comments 5
Minor editing during proof-reading.
Response:
Thank you for your review of the manuscript, we appreciate your valuable feedback. We will try our best to correct the problems in the article in the process of proof-reading.
